# Exploring the Moral Hazard Evolutionary Mechanism for BIM Implementation in an Integrated Project Team

**Yanchao Du [1],\***  **, Hengyu Zhou [1], Yongbo Yuan [1] and Hong Xue [2]**

[1]  Department of Construction Management, Dalian University of Technology, Dalian 116024, China;
   zhy19880607@mail.dlut.edu.cn (H.Z.); yongbo@dlut.edu.cn (Y.Y.)
[2]  School of Management, Harbin Institute of Technology, Harbin 150001, China; xuehong@stu.hit.edu.cn
\*  Correspondence: Duyanchao@mail.dlut.edu.cn

**Abstract:** Integrated project delivery (IPD) is a new emerging delivery system, contributes to increase value to the owner, reduces waste and maximizes efficiency in the life cycle of projects. However, IPD system has not yet shifted from pilot-alike or particular-purposed cases to large-scale applications. The huge advantages of building information modeling (BIM) are far from being exploited, which directly leads to the delivered outcomes below expectations, thereby causing obstacles to widespread application of IPD system. The reasons impeding the successful application of BIM has been a hot topic. Previous studies suggested that moral hazard behavior is a critical inducer leading to the undesirable outcomes. However, very few studies have studied the evolution mechanism of moral hazard behavior for BIM application. To fill this knowledge gap, this study proposed a novel model, aiming to capture dynamically the interactive behavior of BIM-based strategy selections using evolutionary game theory. Five parameters of monitoring cost, proprietary cost, incentive payment, punishment and speculative benefit are extracted and defined in the proposed model. Numerical simulations are conducted with MATLAB 2016a. The simulation results showed that when incentive payment is higher than the sum of speculative benefit and proprietary cost, interactive behavior of both game players will move toward the optimal portfolio strategy. Incentive payment and punishment have negative correlations with the probability of moral hazard behavior for BIM application. Parameters of speculative benefit and proprietary cost affect positively implementation probability of moral hazard behavior of employing BIM. This study can provide theoretical and managerial implications for integrated project managers and related government department to improve implementation of BIM and IPD system, and also contribute to its sustainable development.

**Keywords:** integrated project delivery; building information modeling; moral hazard behavior; evolutionary game model

## 1. Introduction

In recent years, China is embarking the largest development of construction project in the world, and increasingly complex engineering structures and super-large projects spring up like mushrooms, which brings great difficulties to information integration and makes all stakeholders face huge investment and management risks [1]. Under this background, the enormous demand has kindled construction enterprises and scholars widespread attention to building information modeling (BIM) in the architecture, engineering and construction (AEC) industry. However, it is difficult for construction enterprises to implement BIM in traditional project delivery system such as design–bid–build (DBB), design–build (D) and construction management at risk (CMR), which easily lead to information and

process fragmentation, owing to the inherent drawbacks of traditional organization structure [2]. Therefore, the implementation of BIM calls for a new project delivery system.

Integrated project delivery (IPD) is defined as a new emerging project delivery system that integrates people, system, business structure and practices into a process that makes full use of the knowledge and insight possessed by key stakeholders within integrated project team, so as to optimize project performance, increase values to the owner, improve efficiency and effectiveness throughout the lifecycle of projects, and finally achieve project success [3]. All participants of the integrated project team, including the owner, designer, contractor and subcontractors, are connected by the multi-party relationship contract, taking BIM as collaboration platform to transmit architecture, structure, mechanical, electrical and plumbing information. The integrated project team will be established at an early design stage, and work collaboratively by sharing the benefit and risk in the project lifecycle [4]. Enterprises and scholars unanimously believed that BIM and IPD are interdependent and inseparable. BIM provides a perfect collaboration environment for all participants, shifts traditional project management from the two-dimensional to three-dimensional mode, and improves the quality and efficiency of design and construction process [5]. Moreover, BIM contributes to change project management from intra-firm to inter-firm collaboration [6], and is described to be so advanced that it will give rise to a paradigm shift in the AEC industry [7]. The IPD system overcomes the drawbacks of fragmented process and information that always exists in traditional project delivery approaches, and provides continuous information flow and the synergetic process for successful application of BIM [8].

In IPD system, target value design (TVD) is an essential part of the integrated agreement. TVD involves designing to a specific estimate instead of estimating based on a detailed design. It seeks to address the problem that affects the problem that influences many projects—various design disciplines work from a common schematic design to do design development in their areas of expertise. With little cross-functional collaboration, project design cost is often high, resulting in construction projects being unconstructable and delayed. Corrective action may include a misuse of value engineering to radically cut the scope of projects, or to suppress certain features that are desirable but unaffordable. Moreover, the lack of collaboration often results in early design decisions that are later found to be suboptimal, but may be difficult to change. Eventually, much time and effort are wasted, and the design cycle is longer that it should be. These waste run counter to lean construction.

TVD argues that stakeholders in integrated project team should "do it right the first time" and build constructability into their designs and construction, as opposed to design first and then evaluating constructability later.

TVD suggests concurrent design, with various disciplines in ongoing contact, as opposed to periodic reviews.

Solution sets should be performed in the design process to make sure that good alternatives can be available later.

Macomber and Howell [9] proposed a number of foundational practices for target value design, this practice promotes design conversation, as they see design as a social activity that involves several professionals focusing on meeting the needs of the client. This approach is especially effective in light of the fact that the client's needs can change over time and value assessments need to be repeatedly made to ensure that design decision meet these needs. The details are as follows:

1.　Conduct design activities in a big room (Obeya in Japanese). Toyota has used it successfully, especially in product development, to enhance effective and timely communication. The Obeya is similar in concept to traditional war rooms, and will contain charts, pictures and graphs on display boards that visually represent program timing, milestones and progress to date. The boards also display actions or recommendations to resolve delays or technical problems. The Obeya houses project leaders and key staff in close but comfortable proximity to shorten the communication cycle and promote an effective plan, do, check and act (PDCA) cycle. Spontaneity comes easily as specialists can collaborate readily in key design or construction decisions.

2.　Work closely with the client to establish the target value. Designers should guide clients to establish what represents value and how that value is produced. They should ensure that clients are active participants in the process, not passive customers.

3.　Once the target value is established, use it to work with a detailed estimate. Have the design team develop a method for estimating the cost of design alternatives as they are developed. Deviations should not continue unchecked; if a particular design feature exceeds the budget allocated for it, then that design should be adjusted promptly in order not to abort further design work that cannot be accepted.

4.　Apply concurrent design principles to design both the product and the process that will produce it. This work should be done as a collaboration between architects/engineers, specialty designers, contractors/subcontractors and the client. Be flexible to include innovation in this process. Practice reviewing and approving design work as it progresses.

5.　Working in small, diverse groups are best. Groups of eight or fewer people establish better group dynamics; it is easier to create a spirit of collegiality and trust that lead in turn to more innovation and learning. Design with the customer in mind. Focus on designing in the sequence of the discipline that will use it. Use the "pull" approach with each design assignment to serve the next discipline. Lean is obtained by meeting downstream needs as opposed to producing what is convenient. Overproduction increases the possibility that the work so produced may not be what is needed for the next discipline to maintain its schedule, and it may lack the collaboration necessary for constructability.

6.　Collaboratively plan and replan the project. Planning should involve all stakeholders to continually maintain an actionable schedule. Joint planning will refine practices of coordinating action. This will avoid delay, rework and out-of-sequence design.

7.　Lead the design effort for learning and innovation. Expect the team to learn and produce something surprising. Expect also surprise events to upset the current plan and require more re-planning.

8.　Learn by carrying out conversations on the results of each design cycle. Include all project participants in order to capture knowledge on success factors. Use this information as a part of the PDCA cycle, and use formal measurement systems, if possible.

At the beginning of the design, a lot of manpower and material resources are needed to work together with client to determine what represents value and what it might cost. Subsequently, the total budget is allocated to various facility systems and cross-functional teams collaborate to work within budget limits. Budget should be updated quickly to keep pace with design process. This approach varies substantially form traditional practice that is highly linear. In TVD, budget is the influencing factor of design, rather than an outcome of design.

In general, for this process to operate successfully, the integrated project team members who are responsible for construction should develop budget based on preliminary design drawing instead of contract drawing and specifications. Designers need to accept critical comments on their cost and constructability. The owner must be willing to be questioned and to provide feedback on balancing project costs with specific building functions. This may involve value engineering compromises of life-cycle costs versus form, function and time. TVD brings a standard process to the integrated project team, as shown in Figure 1. In addition, TVD can be combined with quality function deployment (QFD) that plays a significant role in capturing the requirements of the owner and quantifying the importance of functions, to find a balance between functions and budget limits. QFD is a relative and complex approach, which requires the integrated project team member with rich experiences, under this situation, it can be implemented successfully. QFD approach is beyond the research scope in this paper, so the authors will not discuss it in detail here.

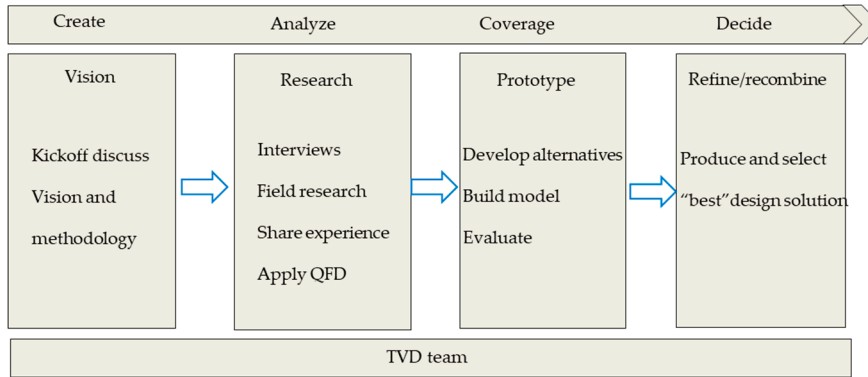

**Figure 1.** The target value design process.

Ballard and Reiser [10] reported firstly a project, named Saint Olaf College Fieldhouse that was built between 2001 and 2002, that successfully took advantage of TVD. This project was delivered on time, within budget and with more value provided to the owner compared to traditional project delivery systems. Meanwhile, the contractor, subcontractors and designers also made generous profits. A similar project that was built between 1998 and 2002, took more than 10 months and the cost would be 18% higher. TVD was also adopted as a significant component of integrated form of agreement (IFOA) for lean delivery in Sutter health project located in Sacramento, CA., which achieved great success [11]. In this project, TVD is intended to have a project designed within a desired budget in line with a detailed estimate, and establishes value, cost, schedule and constructability as basic components of the design standard. Actually, TVD focuses on the objective of creating the best design for a facility that can be delivered with the funding available. Above all, the IPD system that is supported by technologies of lean construction, including TVD, QFD, etc., has a great advantage in improving the entire project performance, in contrast with traditional delivery systems.

Despite huge advantages, the IPD system has not yet shifted from single pilot or particular-purposed cases to large-scale applications. From the perspective of application practice, the capacities of BIM are far from being exploited, which directly leads to the delivered outcomes below expectations, thereby creating great obstacles to widespread application of the IPD system [12]. This presents us with a question of why BIM application is impeded in integrated project team notwithstanding a large amount of advantages described by scholars and experts in extant literatures. According to extant literatures, most researchers have paid too much attention to technology research and development technology factors are no longer the main obstacles to the application of BIM [13]. As a matter of fact, BIM is a sociotechnical system more than technology [14], and the acceptance of BIM significantly influenced by many elements derived from the social attribute of organizations [15]. Many factors that impede BIM application have been studied by researchers such as financial decision-making [16], process and technology [17] and human elements [18]. However, to the best of our knowledge, very few studies have focused on the conflicts between individual benefit maximization and inter-organization collaboration. For the integrated project team, notwithstanding the owner usually plays a crucial role in promoting BIM application, they significantly depend on the other participants' expertise and experience for achieving BIM-enabled project success. Hence, to some extent, the IPD system also complies with the classical model that is known as the principal-agent model. Although the IPD system highlights that stakeholders' interest are aligned with the overall interest of projects, they are still self-interested and try their best to maximize individual utilities because stakeholders are limited by bounded rationality [19]. In that context, stakeholders, including the contractor, designer and subcontractors, would likely behave in an inappropriate way to harm the owner's interest, and thus reduce the overall project performance.

BIM is a collaborative information platform, in which stakeholders should share information with each other. This means that stakeholders' proprietary information is open to the whole integrated

project team, which often results in intense competition among stakeholders and strict supervision from the owner within integrated project team, and thus influence vested benefit possessed by stakeholders. That is to say, to some degree, information sharing behavior possibly brings in losses for stakeholders themselves. As being stipulated in the IPD contract, stakeholders are often required to use BIM in the lifecycle of projects. They would trade off cost and benefit, and are inclined to adopt BIM at a low level effort when confronting an uncertain benefit compared with cost, which may lead to overall project performance below expectations.

To encourage all stakeholders to jointly share benefit and risk, the concept of common-pool is introduced into the IPD system and often is stipulated in multi-party relationship contract. When implementing BIM, stakeholders, however, frequently encounter the conundrum of 'common-pool' resources, which often causes serious consequences because of free-riding behavior [20]. Each participant would neither receive the entire benefit nor take all the responsibility, even though they adopt BIM at high level effort or cause losses at low level effort. As a result, participants are very likely not to cooperate sincerely, and only provide conceptual BIM-based product to the owner. This is a severe problem that existed in IPD-based organizations, which is known as a moral hazard dilemma that is adverse to the overall project performance in the AEC industry [21].

The moral hazard problem has not been paid more attention in the IPD system. As a matter of fact, under the cover of the principle of the "overall goal" and "benefit sharing", researchers often ignore the non-cooperative attitude of project team members in practice. Consequently, not only benefit sharing could not be achieved, but also unnecessary risks and losses are borne. Identifying and effectively preventing moral hazard behavior caused by BIM application has been a great challenge for researchers [22]. Although some researchers has previously mentioned moral hazard in terms of the concept level, how to capture the dynamic interactive behaviors among stakeholders and predict the evolutionary trend in integrated project team, namely the evolutionary mechanism, is rarely discussed in extant literatures. Therefore, the objective of this paper is to explore BIM-base selection strategies and reveal the evolutionary mechanism for the purpose of developing preventive measures to control moral hazard. This study has both theoretical and practical contributions. Firstly, we fill the knowledge gap that the inherent evolutionary mechanism behind BIM-based selection strategies is lacking in current research. Secondly, this study can provide significant and valuable implications for project managers and related government department to take effective measures to control moral hazard behavior in an integrated project team.

The rest of this study is arranged as follows: In Section 2, a systematic literature review is presented. In Section 3, the reason for choosing evolutionary game model is elaborated. In Section 4, the proposed novel model is established by means of payoff matrix and the replicator system. In Section 5, this study conducts numerical simulations to explore the evolutionary trend of combined strategies adopted by two game players, and probe into the impact of model variables on evolutionary trend of interactive behavior under constraint conditions. In Section 6, the simulation results are obtained. In Section 7, conclusions and implications are given.

## 2. Literature Review

The concept of moral hazard is first proposed by Arrow [23] in the study of insurance contracts, which originally points out that the persons purchasing insurance, afterwards they begin to be careless to prevent risk, even making an accident maliciously, and thus attempts to acquire insurance income. On that background, the policy of insurance would deviate from its original incentive direction and change the probability of insured accidents, so that a fire insurance policy with an excess value over an insured item may induce arson attack or deficiency management that is caused deliberately by the insured persons. With the deepening of research, scholars have expanded the connotation of moral hazard, and defined it as people who engage in economic activities acting against someone else while maximizing individual utility [24]. Moral hazard dilemma generally exists in situations

where, owing to uncertainty and incomplete contracts, the team members do not necessarily bear or enjoy all the losses or benefit caused by their own actions [25].

In the field of information economics, many researchers have earlier explored moral hazard with the principal-agent model that is based on asymmetric information game theory. For instance, Rubbinstein [26] and Radner [27], by using the repeated game model, proved that the effect of risk sharing and the incentive mechanism can be achieved with the first order optimal condition of pareto, under the assumption that the client and agent maintains a long-term relationship. The repeated game model is established based on the classical game theory that emphasizes that the people are absolutely rational and holds complete information on the competitors [28]. However, this extremely strict assumption cannot be satisfied in practice because of game players' knowledge and time limitation [16]. The principle usually relies on the information sent by the agent, and thus results in asymmetric information between agent and principal. Due to the asymmetry of information, the principle cannot obtain complete information, but needs to assume full responsibility, while the agent, by virtue of his own advantages, is likely to conceal information and lower the level of effort without undertaking negative consequences caused by itself [29]. In general, researchers in the field of information economics is to analyze the moral hazard problem by establishing a game model under given information status quo, and then design corresponding incentive and restraint mechanisms to curb the agent's moral hazard behavior. However, it overemphasizes the "economic man" characteristics of agents, and fails to consider the characteristics of bounded rationality and incomplete information, which has great limitations in practice. New institutional economics argued that moral hazard is team members' opportunistic behavior in essence [30]. They proposed a modification to the hypothesis of "economic man" that existed in information economics, and thus put forward three assumptions, namely utility maximization, bounded rationality and opportunism for exploring moral hazard behavior [31].

With the increase of global competitiveness, researchers and construction practitioners have been continually seeking to apply better technologies and processes to improve project delivery, but there is a lack of unified strategy and there is little incentive to change. Most construction contracts place the parties to construction in adversarial roles, although delivery systems such as DB and DBB systems have diminished this challenge to a limited extent. The core of lean construction is the objective of "global optimization", in which overall project performance is maximized, compared with the "local optimization" where individual stakeholder's benefit usually is at the expense of others. The fundamental principle of lean construction is to reduce or eliminate waste, BIM addresses mangy aspects of waste that occur in design and construction stages. Essentially, lean construction and BIM supplement each other, and their compatibility is the basis of collaboration between critical technologies of lean construction and BIM.

Firstly, the implementation of lean construction can create a good environment for BIM application. After more than 20 years of development, lean construction has formed a relatively mature theoretical system. The lean construction management mode focuses on client's requirements, always aiming at maximizing the value of client, constantly optimizing and improving the workflow by applying value engineering and emphasizing the participation of all project team members. The importance of information management cannot be overemphasized in lean construction. In comparison with the traditional delivery system, lean construction can create a more efficient environment for information transmission, which also provides an important condition for BIM implementation. Secondly, BIM could improve the implementation effect of lean construction. Building information modeling (BIM) is the process of generating and managing building data during its life cycle. It is also a tool as well as a process, and increases productivity and accuracy in the design and construction of buildings. BIM uses 3-D, dynamic building modeling software and operates in real time. It supports the continuous and immediate availability of project design scope, schedule and cost information that is high quality, reliable, integrated and fully coordinated. The building information model encompasses building geometry, spatial relationships, geographic information and quantities and properties of building

components. Though it is not itself a technology, it is supported to varying degrees by different technologies. The BIM is a data-rich, object-oriented, intelligent and parametric digital representation of the facility, from which views and data appropriate to various users' needs can be extracted and analyzed. It generates information that can be used to make decisions and to improve the process of delivering the facility. BIM supports many initiatives that are critical to lean design and construction, so that project participants can timely and accurately obtain project-related information, which can effectively solve the problem of information fragmentation in the construction process, and significantly promotes the implementation effect of lean construction. The IPD system fully absorbs the idea of lean construction, and offers a favorable external environment for successful implementation of BIM. Meanwhile, BIM provides a new technology and process for the realization of the IPD system.

The tremendous demand in the AEC industry has inspired enterprises to invest a large amount of manpower and material resources to adopt BIM in large-scale projects [5]. However, with the deepening of BIM application, the traditional project delivery modes are increasingly restricting the productivity of AEC industry [32]. IPD as an emerging project delivery system has attracted global researchers' widespread attention, because it is believed to play a significant role in reducing costs, integrating fragmented process and information, eliminating risks and promoting the in-depth application of BIM [4]. Matthews and Howell [9] discovered 10% cost-savings by investigating the IPD-based project located in Orlando. Coincidentally, a research presented in the 2006 American Institute Architects (AIA) integrated practice international conference, reported that the IPD system had a bright future, by tracking 40 Australian IPD-enabled projects.

In recent years, however, an increasing number of obstacles to BIM implementation in the IPD system are reported in existing literature [33]. Researchers have started to focus on this issue and conducted a series of studies. Wang et al. [34], on the basis of analyzing the incentive characteristics of traditional project delivery approaches, developed a new incentive model to control the moral hazard behavior of BIM application by non-owner participants, and manifested that the non-owner participants' enthusiasm for using BIM can be mobilized greatly, and they will eventually deliver developed BIM-based products with high quality. However, the mathematic model does not consider the influence of speculative benefit and proprietary cost incurred by the non-owner participants on the selected behaviors of integrated project team members. Proprietary cost actually refers to additional cost paid by the non-owner participants working at high level effort, in comparison with the cost at low level effort [35]. Hence, proprietary cost is a function of effort level, and improves with the increase of effort level. Speculative benefit is regarded as extraneous income acquired by the non-owner participant if they invest these BIM-based resources in other fields or projects other than the current project [36]. Su et al. [37] constructed a multi-incentive model based on the extant literature to probe into the influence of different factors on free-riding behavior. Combined with literature and real cases, this paper used the game simulation method to analyze the impact of project optimization potential, project goal setting, incentive structure and participants' effort cost on the project incentive effect of adopting BIM by the non-owner participants under the concept of the IPD system, the results showed that these four factors have an important impact on the incentive effect. The difference of effort cost among participants is conductive to the success of project, and the benefit distribution should avoid excessive disparity. Nevertheless, the penalty factor is not discussed in this study, which also has an important influence on the selected behavior of non-owner participants. Moreover, game simulation can only explore the influences of these factors on the incentive effect from the static perspective, and cannot dynamically capture the interactive BIM-based behavior among integrated project team members in the lifecycle of projects. Mei et al. [38] studied the rent-seeking behavior between the supervision department and the contractor in BIM-and IPD-based projects by the established game model in which parameters are obtained through a questionnaire survey. The results showed supervision efficiency, incentive and punishment have a very important impact on participants in the integrated project team and manifested that BIM and IPD could effectively control collusive behavior. However,

the data acquired through a questionnaire survey is greatly subjective and uncertain, easily resulting in an unreliable conclusion that needs further investigation.

The systematic review of existing studies shows us such a big picture that IPD, as an advanced project delivery system, along with BIM provides more possibility to improve the project performance. With the project becoming increasingly complex, the IPD system encounters a great challenge to widespread application. More and more researchers have paid attention to this dilemma and explore solutions from different perspectives. Scholars unanimously have recognized the important role of BIM in improving IPD-base project benefit. However very few study has focused on the BIM-based selection strategies from the perspective of moral hazard and tracked how these interactive behaviors evolve dynamically in the integrated project team. Hence, the goal of this paper was to explore the evolutionary mechanism of BIM-based strategy selection, and thus develop corresponding measures to control moral hazard behavior.

## 3. Methodology

Intense competition and cooperation exists among stakeholders in the BIM-and IPD-based project. The interactive behavior of strategy selection related to BIM application is a complicated and dynamic process. The evolutionary game model has been successfully applied in the economy and society to analyze the long-term economic and transaction behaviors [39]. It relaxes the restriction of absolute rationality, namely the hypothesis of the "Economic man" that is emphasized repeatedly in the asymmetric information game theory, because it is impossible to require each player to fully learn about the competitor's information at the beginning. As a matter of fact, each player acquires the competitor's information through constant trial and error, learning, imitation and correction behavior. The evolutionary game model replaced absolute rationality with bounded rationality implying that the integrated project team members' cognitive abilities are limited, and they cannot completely get hold of the competitor's information [19]. The evolutionary game model that integrates the game analysis and evolutionary dynamics, mainly handles the complicated problems related with diffusing and propagating behavior, and is regarded as a highly effective method to analyze the interactive behavior of cooperation and competition [40]. The replicator equation and evolutionary game stable state (ESS) as two core concepts of the evolutionary game model, can be taken advantage of dynamically capturing the interactive behavior and describing the evolution state quo [28], and thus predicting outcomes of the integrated project team performance. More importantly, we could simulate game players' behavior as close to reality as possible when the case data is difficult to obtain, which would provide the project manager and related government department actionable insights in the future.

Based on the analyzation mentioned above, we could see that evolutionary game model is a reliable approach to track the dynamic and interactive behavior of BIM-based strategy selection. Therefore, we developed a novel evolutionary game model considering incentive and punishment to probe into the evolutionary mechanism of moral hazard for successful implementation of BIM in an integrated project team.

## 4. Evolutionary Game Model

### 4.1. Model Parameters

In terms of previous systematic research review on moral hazard in organization management, we confirmed several factors of monitoring cost, incentive, punishment, speculative benefit and proprietary costs, which have been rarely explored systematically in the moral hazard behavior evolutionary mechanism. These elements significantly play a very important role in BIM-based strategies selection behaviors in an integrated project team. However, how these strategies interact dynamically in the evolutionary process is still unknown under these factors. Thus these factors were extracted as variables of the evolutionary game model and presented as Table 1.

**Table 1.** Variables and meanings.

| No. | Variables | Content |
| --- | --- | --- |
| 1 | $\xi$ | Refers to the speculative benefit that is interpreted as reduced cost in comparison with the cost at high level effort, when the non-owner participant adopt BIM-based selection strategy at low level effort |
| 2 | $p$ | Refers to the penalty imposed by the owner when the non-owner participant adopt BIM-based selection strategy at low level effort |
| 3 | $\delta$ | Refers to the incentive provided by the owner when the non-owner participant adopt BIM-based selection strategy at high level effort |
| 4 | $c_1$ | Refers to the monitoring cost incurred by the owner |
| 5 | $c_2$ | Refers to the proprietary costs if the non-owner participants adopt a BIM-based selection strategy at high level effort in comparison with cost at low level effort |

*4.2. Hypothesis*

**H1.** *The evolutionary game model is assumed to be an unobservable system including two game players: Player 1 and player 2.*

Firstly, despite that the integrated project team consists of multi-disciplinary stakeholders including the owner, designer, contractor, subcontractor, etc., for the purpose of investigating the interactive behavior of BIM-based selection strategies more precisely, the stakeholders were divided into two types, namely the owner and non-owner stakeholders. Player 1 represents the owner including, and player 2 represents one party randomly selected from the non-owner stakeholders.

Secondly, because of bounded rationality restriction, two players can only grasp partial information on the opposite side before making a decision simultaneously, and both parties cannot observe the opposite side's selection strategy and payoff in every single round game until that round game is over. This means that each player would experience a process of trail and error, learning and corrective action. Ultimately, they would select the optimal BIM-based strategy to maximize their benefit at the end.

**H2.** *Two pure strategies provided for player 1 and 2 is set to be "$A_1$ strategy" and "$A_2$ strategy" and "$B_1$ strategy" and "$B_2$ strategy " respectively, with the probability of x/1-x and y/1-y.*

$A_1$ strategy represents that player 1, to improve the overall project performance, would seriously evaluate and supervise the process of BIM application implemented by player 2 when finding BIM has not worked as well as had been expected. In contrast, player 1 would lose supervision on the BIM application and exploration, namely $A_2$ strategy, when finding BIM applications have not worked as well as had been expected. In consideration of game players' bounded rationalities, which strategy to employ for every single round of games can be regarded as a probability function. Therefore, we assumed that the probability functions for two players to adopt $A_1$ and $B_1$ strategies are x and y, and thus 1-x/1-y for $A_2$/$B_2$ strategies

**H3.** *The revenue gained is assumed to be $l_i$, of which i = (1,2), when player 1 does not take any preventive measures and player 2 works based on the initial expectations.*

**H4.** *Player 2 cannot receive incentive payment if adopting BIM at low level effort. On the contrary, they would get incentive payment δ, which is identified as project cost that is included into the total cost of player 1.*

**H5.** *Player 2 should assume corresponding punishment of p under such circumstances that they adopt BIM at low level effort discovered by player 1, and the punishment is identified as revenue for player 1.*

**H6.** *The speculative benefit ξ for player 2 equals the loss for player 1.*

Based on the hypotheses described above, the payoff matrix is shown in Table 2. Payoff matrix is also referred to as the "winning matrix" that is usually employed to depict the game players' pure strategies and payoff in different combined strategies. As described in Table 1, for instance, the payoffs

of $(A_1, B_1)$ strategy for player 1 and player 2 are $l_1 - \xi - c_1 + p$ and $l_2 + \xi - p$ respectively. The payoff matrix plays an indispensable role in establishing replicator dynamic equation that helps us capture dynamically the interactive behavior between game players, and predict the evolution outcomes.

**Table 2.** Payoff matrix.

| Player 1 | Player 2 | |
|---|---|---|
| | $B_1(\mathbf{y})$ | $B_2(1-\mathbf{y})$ |
| $A_1(x)$ | $l_1 - \xi - c_1 + p, l_2 + \xi - p$ | $l_1 - c_1 - \delta, l_2 + \delta - c_2$ |
| $A_2(1\text{-}x)$ | $l_1 - \xi, l_2 + \xi$ | $l_1 - \delta, l_2 + \delta - c_2$ |

### 4.3. Evolutionary Game Model

We make $\tau^1 = (1, 0)$ express the status where either player adopts the strategy ($A_1$ or $B_1$) with the probability of 1. In a similar vein, define $\tau^2 = (1, 0)$ to express the status where either player selects the strategy ($A_2$ or $B_2$) with the probability of 1.

For player 1, the expected revenue to adopt $A_1$ strategy is set to $f^{A_1}(\tau^1, s)$:

$$
\begin{aligned}
f^{A_1}(\tau^1, s) &= y(l_1 - \xi - c_1 + p) + (1 - y)(l_1 - c_1 - \delta) \\
&= (p - \xi + \delta)y + l_1 - c_1 - \delta
\end{aligned}
\tag{1}
$$

The expected revenue to choose $A_2$ strategy is set to $f^{A_2}(\tau^2, s)$:

$$
\begin{aligned}
f^{A_2}(\tau^2, s) &= y(l_1 - \xi) + (1 - y)l_1 \\
&= l_1 - y\xi
\end{aligned}
\tag{2}
$$

The average expected revenue for player 1 is set to $f(x, s)$:

$$
\begin{aligned}
f(x, s) &= x f^{A_1}(\tau^1, s) + (1 - x) f^{A_2}(\tau^2, s) \\
&= xyp + xy\delta - c_1 x - \delta x + l_1 - y\xi
\end{aligned}
\tag{3}
$$

Based on the evolutionary game theory, if the revenue of a particular strategy is over the average expected revenue, it would spread rapidly among game players and eventually becomes the dominant strategy. Replicator dynamic equation is taken advantage of describing the dynamic evolution trend of the proportion of strategies among game players. For player 1, based on Equations (1) and (3) the replicator dynamic equation is presented as follows:

$$
\eta(x) = \frac{dx}{dt} = x[f^{A_1}(\tau^1, s) - f(x, s)] = x(1 - x)(yp - c_1).
\tag{4}
$$

In a similar way, for player 2, there exists:

$$
\begin{aligned}
f^{B_1}(\tau^1, s) &= x(l_2 + \xi - p) + (1 - x)(l_2 + \xi) \\
&= l_2 + \xi - px
\end{aligned}
\tag{5}
$$

$$
\begin{aligned}
f^{B_2}(\tau^2, s) &= x(l_2 + \delta - c_2) + (1 - x)(l_2 + \delta - c_2) \\
&= l_2 + \delta - c_2
\end{aligned}
\tag{6}
$$

$$
\begin{aligned}
f(y, s) &= y f^{B_1}(\tau^1, s) + (1 - y) f^{B_2}(\tau^2, s) \\
&= y(l_2 + \xi - px) + (1 - y)(l_2 + \delta - c_2)
\end{aligned}
\tag{7}
$$

In the light of Equations (5) and (7), the replicator dynamic equation is presented as follows:

$$
\begin{aligned}
\mu(y) &= \frac{dy}{dt} = y[f^{B_1}(\tau^1, s) - f(y, s)] \\
&= y(1 - y)(\xi - \delta + c_2 - xp)
\end{aligned}
\tag{8}
$$

Combine Equations (4) and (8), a simultaneous equation that is a two-dimensional nonlinear dynamic system, can be obtained as follows:

$$\begin{cases} \frac{dx}{dt} = x[f^{A_1}(\tau^1, s) - f(x, s)] = x(1-x)(yp - c_1) = 0 \\ \frac{dy}{dt} = y[f^{B_1}(\tau^1, s) - f(y, s)] = y(1-y)(\xi - \delta + c_2 - xp) = 0 \end{cases}. \tag{9}$$

By solving the simultaneous equation (9), five solutions that also are known as local equilibrium points (LEP) are obtained as follows: $A(0,0)$, $B(0,1)$, $C(1,0)$, $D(1,1)$ and $E(x^*, y^*)$, wherein:

$$\begin{cases} x^* = \frac{\xi - \delta + c_2}{p} \\ y^* = \frac{c_1}{p} \end{cases}. \tag{10}$$

The Jacobian matrix (J) that plays a critical role in analyzing the evolutionary game stable state (ESS) is as follows:

$$J = \begin{bmatrix} \frac{\partial \eta(x)}{\partial x} & \frac{\partial \eta(x)}{\partial y} \\ \frac{\partial \mu(y)}{\partial x} & \frac{\partial \mu(y)}{\partial y} \end{bmatrix} = \begin{bmatrix} (1-2x)(yp-c) & xp(1-x) \\ yp(y-1) & (1-2y)(\xi - \delta + c_2 - xp) \end{bmatrix}. \tag{11}$$

According to Friedman theory, it is inferred that points of A, B, C and D are LEPs with pure strategy respectively, and the point of E is a LEP with mixed strategy. Through analyzing the values of the determinant and trace of J, namely det(J) and tr(J), we can judge the evolutionary status for every LEP. The results of det(J) and tr(J) for each point are calculated and shown in Table 3.

**Table 3.** Determinant and trace of Jacobian.

| LEP | The Determinant and Trace of J |
|---|---|
| A(0,0) | det(J) $= (\delta - c_2 - \xi) * c_1$ <br> tr(J) $= \xi - c_1 - \delta + c_2$ |
| B(0,1) | det(J) $= (p - c_1)(\delta - c_2 - \xi)$ <br> tr(J) $= p + \delta - c_2 - c_1 - \xi$ |
| C(1,0) | det(J) $= c_1 * (\xi - \delta + c_2 - p)$ <br> tr(J) $= \xi + c_1 - \delta + c_2 - p$ |
| D(1,1) | det(J) $= (p - c_1)(\xi - \delta + c_2 - p)$ <br> tr(J) $= \delta - c_2 + c_1 - \xi$ |
| E(x*,y*) | det(J) $= \frac{c_1 * (\delta - c_2 - \xi)(p + \delta - c_2 - \xi)(c_1 - p)}{p^2}$ <br> tr(J) $= 0$ |

For each LEP, first of all, if it complies with the following condition that $det(J) > 0$ and $tr(J) < 0$, then it can be judged as ESS and the combined strategy it represents belongs to the evolutionary stability strategy. Secondly, provided that it satisfies the constraint condition $det(J) \geq 0$ and $tr(J) > 0$, it can be regarded as an unstable point, the combined strategy it represents is unstable. Thirdly, if $det(J) > 0$ and $tr(J) = 0$, it is neutral. Furthermore, provided that it meets the condition $det(J) < 0$, the LEP is judged as a saddle point. According to the analysis above, the results are presented in Table 4.

**Table 4.** Classification and analysis results of local equilibrium points (LEPs).

| Scenarios | Range of Values | LEP | det(J) | tr(J) | Equilibrium Results |
|---|---|---|---|---|---|
| Scenario 1 | $\delta - c_2 > \xi, p > c_1$ | (0,0) | + | − | ESS |
| | | (0,1) | + | + | Unstable |
| | | (1,0) | − | − | Saddle |
| | | (1,1) | − | + | Saddle |
| | | (x*,y*) | − | 0 | Saddle |

**Table 4.** *Cont.*

| Scenarios | Range of Values | LEP | det(J) | tr(J) | Equilibrium Results |
|---|---|---|---|---|---|
| Scenario 2 | $\delta - c_2 > \xi, p < c_1$ | (0,0) | + | − | ESS |
| | | (0,1) | − | Uncertain | Saddle |
| | | (1,0) | − | Uncertain | Saddle |
| | | (1,1) | + | + | Saddle |
| | | (x*,y*) | + | 0 | Neutral |
| Scenario 3 | $\delta - c_2 < \xi, c_1 < p < \xi + c_2 - \delta$ | (0,0) | − | Uncertain | Saddle |
| | | (0,1) | − | Uncertain | Saddle |
| | | (1,0) | + | + | Unstable |
| | | (1,1) | + | − | ESS |
| | | (x*,y*) | − | 0 | Saddle |
| Scenario 4 | $\delta - c_2 < \xi, p > c_1$ and $p > \xi + c_2 - \delta$ | (0,0) | − | Uncertain | Saddle |
| | | (0,1) | − | Uncertain | Saddle |
| | | (1,0) | − | Uncertain | Saddle |
| | | (1,1) | − | Uncertain | Saddle |
| | | (x*,y*) | + | 0 | Neutral |
| Scenario 5 | $\delta - c_2 < \xi, p < c_1$ and $p < \xi + c_2 - \delta$ | (0,0) | − | Uncertain | Saddle |
| | | (0,1) | + | − | ESS |
| | | (1,0) | + | + | Unstable |
| | | (1,1) | − | Uncertain | Saddle |
| | | (x*,y*) | + | 0 | Neutral |
| Scenario 6 | $\delta - c_2 < \xi, \xi + c_2 - \delta < p < c_1$ | (0,0) | − | Uncertain | Saddle |
| | | (0,1) | + | − | ESS |
| | | (1,0) | − | + | Saddle |
| | | (1,1) | + | + | Unstable |
| | | (x*,y*) | − | 0 | Saddle |

## 5. Simulation of the Evolutionary Game Model

To track the evolution process of BIM-based interactive behavior between the owner and the non-owner participant and verify the reliability of evolutionary results described in the previous section, the authors, based on the MATLAB 2016a platform, make full use of the Runge-Kutta equation to dynamically and authentically simulate both parties' interactive behavior in multi-round games.

Function ode45 is called from the MATLAB 2016a platform to acquire evolutionary paths of dynamic interactive behavior. Original values should be set before the simulation experiment. For six scenarios, we make the following assumptions that the initial probability of combined strategies is defined as $(x_0, y_0)$, wherein $0 \leq x_0, y_0 \leq 1$. Based on classification and analysis in Section 4, the baseline values of model parameters $c_1, c_2, \delta, \xi$ and $p$ are assigned and presented respectively in Figure 2.

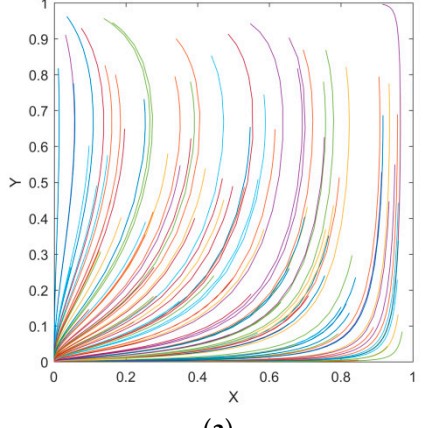

(a)

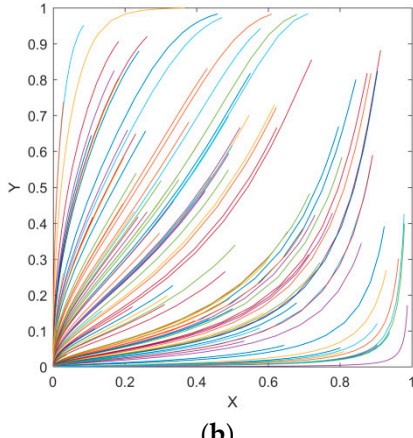

(b)

**Figure 2.** *Cont.*

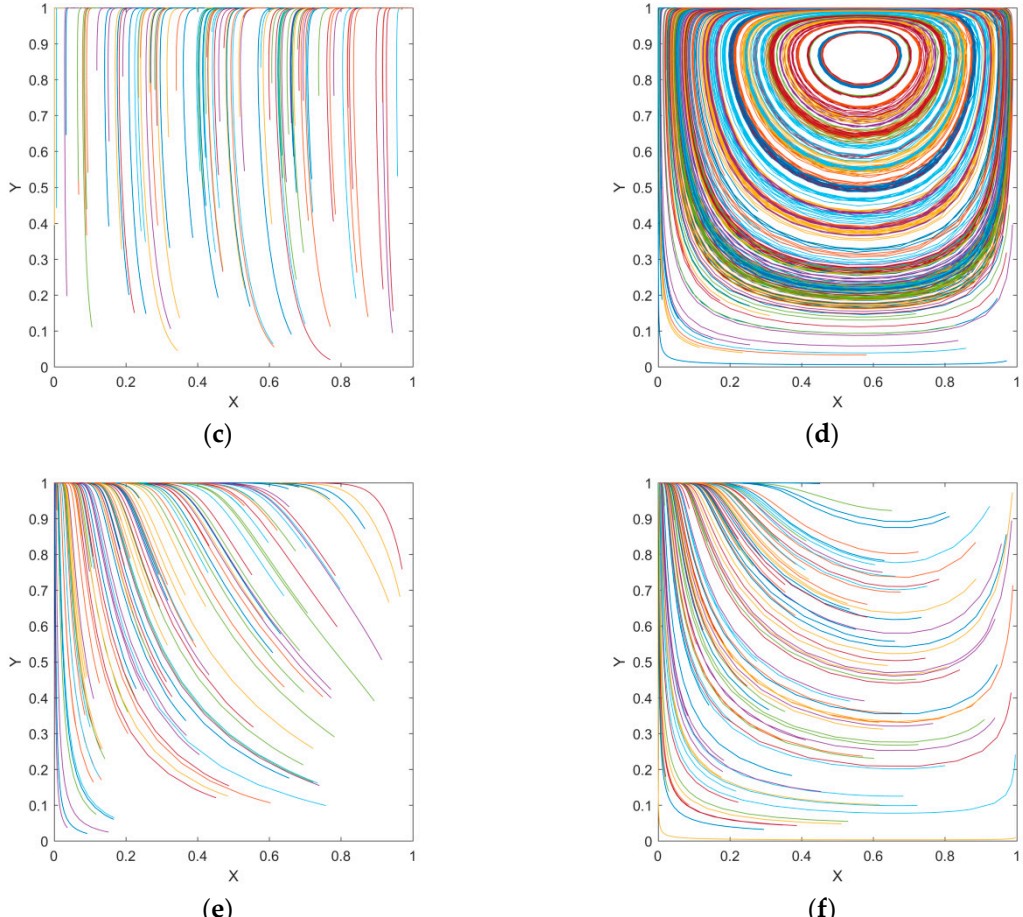

**Figure 2.** Evolution paths: (**a**) Scenario 1: $c_1 = 40{,}000$, $c_2 = 10{,}000$, $\delta = 60{,}000$, $\xi = 30{,}000$ and $p = 60{,}000$; (**b**) scenario 2: $c_1 = 40{,}000$, $c_2 = 10{,}000$, $\delta = 60{,}000$, $\xi = 30{,}000$ and $p = 32{,}000$; (**c**) scenario 3: $c_1 = 10{,}000$, $c_2 = 10{,}000$, $\delta = 13{,}000$, $\xi = 58{,}000$ and $p = 12{,}000$; (**d**) scenario 4: $c_1 = 20{,}000$, $c_2 = 2000$, $\delta = 15{,}000$, $\xi = 26{,}000$ and $p = 23{,}000$; (**e**) scenario 5: $c_1 = 20{,}000$, $c_2 = 2000$, $\delta = 15{,}000$, $\xi = 36{,}000$ and $p = 18{,}000$ and (**f**) scenario 6: $c_1 = 32{,}000$, $c_2 = 1000$, $\delta = 12{,}000$, $\xi = 31{,}000$ and $p = 30{,}000$.

## 5.1. Simulation of BIM-Based Interactive Behavior

In scenario 1, to simulate the evolutionary trend of interactive behavior between two players, we chose randomly 100 array points as original points that symbolize the probability of BIM-based combined strategy adopted by both players respectively under the constraint condition $\delta - c_2 > \xi$ and $p > c_1$. The simulation results, as shown in Figure 1a, demonstrate that the evolution paths were attracted to the local equilibrium point $(0, 0)$. That is to say, the non-owner participant would try their best to implement BIM for the purpose of delivering required three-dimensional digital products with high quality, and the owner would choose the strategy of losing supervision on BIM application and exploration. In term of simulation results, we could draw a conclusion that whatever initial combined strategies adopted by both players, they would choose ESS of $(A_2, B_2)$ eventually, when the incentive is greater than the sum of speculative benefit and proprietary cost incurred by the non-owner participant, meanwhile the punishment is greater than the monitoring cost incurred by the owner.

In scenario 2, to simulate the evolutionary trend of interactive behavior between two players, 100 array points were picked randomly under the constraint condition $\delta - c_2 > \xi$ and $p < c_1$. The evolution results, as shown in Figure 1b, reveal that the evolution paths converged fast to the local equilibrium point $(0, 0)$, namely ESS of $(A_2, B_2)$, The simulation results implied that the non-owner participant would focus all of their energy on implementing BIM so as to deliver the required

three-dimensional digital products with high quality, and the owner would choose to losing supervision on BIM application and exploration eventually, under the constraint condition mentioned above.

In scenario 3, to simulate the evolutionary trend of interactive behavior between two players, we selected randomly 100 array points under the constraint condition $\delta - c_2 < \xi$ and $c_1 < p < \xi + c_2 - \delta$. The evolution paths, as presented in Figure 1c, manifest that no matter what original combined strategies employed by two players, all combined strategies converged to the local equilibrium point (1,1), namely ESS of $(A_1, B_1)$. In this context, the non-owner participant would apply BIM at low level effort to just deliver a conceptual BIM products even though assuming punishment imposed by the owner, while the owner would strengthen supervision and strictly stipulated the scope of BIM application and detailed content of the work.

In scenario 4, 100 array points were picked randomly to explore the evolution trend under the restricted condition $\delta - c_2 < \xi, p > c_1$ and $p > \xi + c_2 - \delta$, According to the evolution paths as shown in Figure 1d, it can be inferred that both players were constantly changing their combined strategies around the of $(\frac{\xi - \delta + c_2}{p}, \frac{c_1}{p})$, which implies that they could not achieve a stable equilibrium strategy.

In scenario 5, 100 array points were picked randomly to explore the evolution trend under the restricted condition $\delta - c_2 < \xi, p < c_1$ and $p < \xi + c_2 - \delta$. According to the simulation results as presented in Figure 1e, whatever initial combined strategies employed by both players, all of them would evolve to the local equilibrium point of (0,1), namely ESS of $(A_2, B_1)$. In this context, the non-owner participant would put BIM into use at low level effort even though they assume the punishment imposed by the owner. Differing from the simulation results in scenario 4, the owner would adopt the strategy of losing supervision on BIM application exploration, because the losses far outweighing the gains if strengthening supervision on BIM application.

In scenario 6, 100 array points are picked randomly to explore the evolution trend under the constraint condition $\delta - c_2 < \xi$ and $\xi + c_2 - \delta < p < c_1$. According to the simulation results, as shown in Figure 2f, whatever initial combined strategies employed by both players, they would evolve to the local equilibrium point of (0,1), namely ESS of $(A_2, B_1)$. In this context, the owner would use the strategy of losing supervision on BIM application and exploration, and the non-owner participant would still put BIM into use at low level effort even though they assume the punishment imposed by the owner.

*5.2. Impact of Model Variables on the Evolution Trend*

5.2.1. Impact of Model Variables on the Evolution Trend in Scenarios 1 and 2

In scenario 1, we first explored the potential impact of monitoring cost ($c_1$) on the strategies selection adopted by both players. The starting point was assigned the value of (0.4, 0.6). The parameter $c_1$ ranged from 10,000 to 50,000 with a step size of 10,000, leaving all the other parameters at their default values. According to numerical simulation results, as shown in Figure 3a, we found that with the increase of supervision cost, the probability of adopting $A_1$ strategy by the owner decreased rapidly to zero, which means that player 1 would likely abandon strict supervision on BIM application by player 2.

Then, we analyzed the influence of proprietary costs ($c_2$) on the evolution trend. The starting point was assigned the value of (0.4, 0.6). The parameter $c_2$ ranged from 5000 to 25,000 with a step size of 5000, leaving all the other parameters at their default values. On the basis of numerical simulation outcomes as presented in Figure 3b, we found that the smaller proprietary cost, the faster player 2 adopted the $B_2$ strategy, namely the stronger the willingness of delivering BIM-based products with high quality to the owner.

Next, we explored the impact of the incentive ($\delta$) on evolutionary direction. The initial point was assigned the value of (0.4, 0.6). The parameter $\delta$ ranged from 50,000 to 90,000 with a step size of 10,000, leaving all the other parameters at their default values. According to the numerical simulation results presented in Figure 3c, the larger the incentive was, the faster the probability of employing $B_2$ strategy

by player 2 converged to 1. Thus parameter $\delta$ played a significant role in improving enthusiasm of the non-owner participant to use BIM at high level effort in practice.

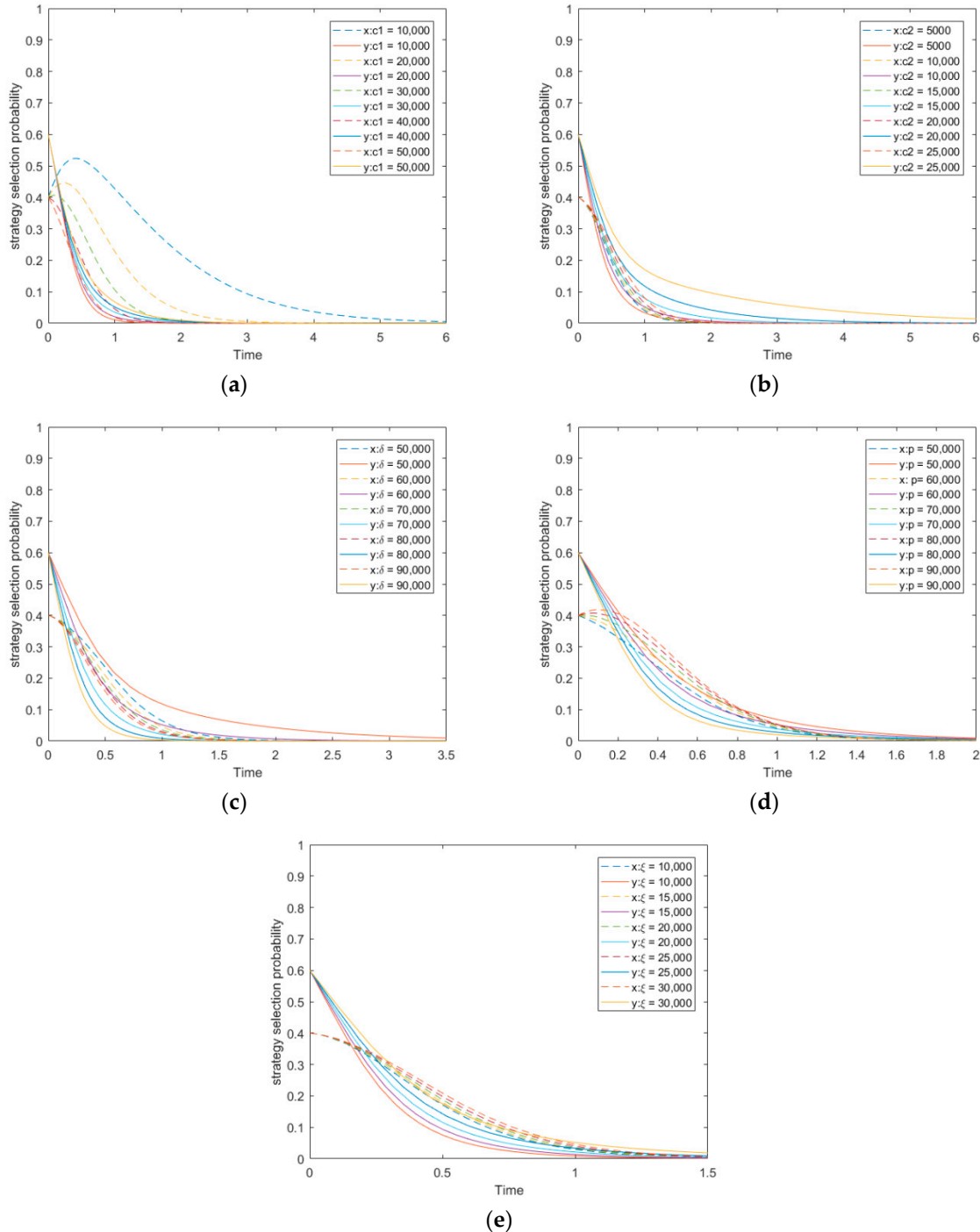

**Figure 3.** Impact of model variables on the evolution trend in scenarios 1. (**a**) Impact of parameter $c_1$, wherein $c_2 = 10,000$, $\delta = 60,000$, $\xi = 30,000$ and $p = 60,000$; (**b**) impact of parameter $c_2$, wherein $c_1 = 40,000$, $\delta = 60,000$, $\xi = 30,000$ and $p = 60,000$; (c) impact of parameter $\delta$, wherein $c_1 = 40,000$, $c_2 = 10,000$, $\xi = 30,000$ and $p = 60,000$; (**d**) impact of parameter $p$, wherein $c_1 = 40,000$, $c_2 = 10,000$, $\delta = 60,000$ and $\xi = 30,000$ and (**e**) impact of parameter $\xi$, wherein $c_1 = 40,000$, $c_2 = 10,000$, $\delta = 60,000$ and $p = 60,000$.

Similarly, we analyzed the impact of the punishment ($p$) on the evolutionary trend. The starting point was assigned the value of (0.4, 0.6). The parameter $p$ ranged from 50,000 to 90,000 with a step

length of 10,000, leaving all the other parameters at their default values. Based on the numerical simulation outcomes, as shown in Figure 3d, we found that the punishment imposed by the owner could effectively control moral hazard behavior from the non-owner participant.

Keeping the other parameters unchanged, the speculative benefit ($\xi$) ranged from 10,000 to 30,000 with a step length of 5000. The initial point was assigned the value of (0.4, 0.6). Based on the numerical simulation results, as shown in Figure 3e, we found that when speculative benefit obtained by the non-participant increased, the willingness to apply BIM at high level effort slowed down, which reveals a profound truth that speculative benefit gained by the non-participant is a key element that causes moral hazard. Additionally, the increase of parameter $\xi$ impeded the owner's endeavor of carefully evaluating and supervision on BIM application, because the owner with the characteristic of bounded rationality believed that if the speculative benefit reached a certain level, the non-owner participant behavior was almost unlimited by the incentive and punishment. From the perspective of the owner, over-regulation possibly caused high cost, endangering individual benefit.

In scenario 2, the initial points were uniformly set to (0.4, 0.6), leaving all the other parameters unchanged when exploring the influence of one particular parameter on the evolutionary trend. The value ranges of five parameters that need to be analyzed in detail are presented at the upper right corner of Figure 4a–e respectively. Based on the numerical simulation results, the influences of five factors were in accordance with the corresponding simulation results in scenario 1. Therefore, the authors would not explain them more here.

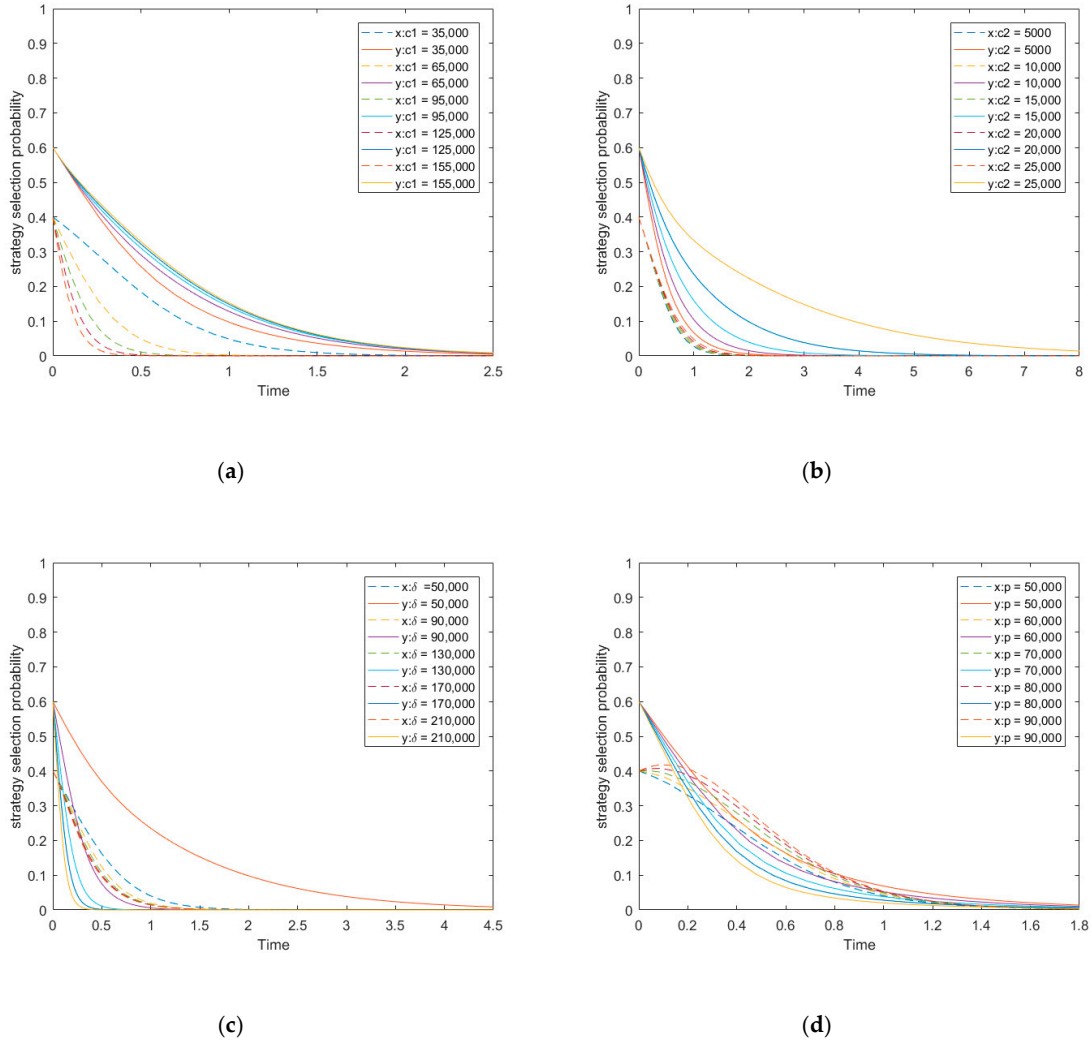

**Figure 4.** *Cont.*

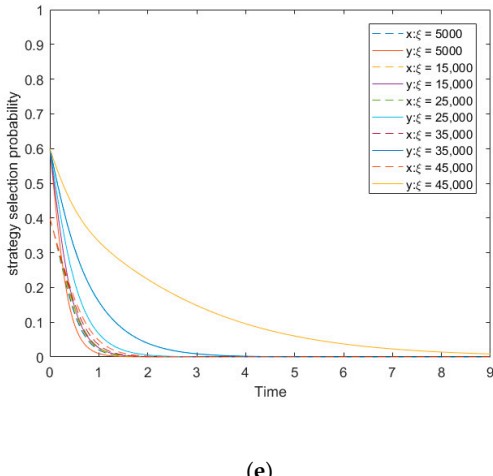

(**e**)

**Figure 4.** Impact of model variables on evolution trend in scenarios 2. (**a**) Impact of parameter $c_1$, wherein $c_2 = 10{,}000$, $\delta = 60{,}000$, $\xi = 30{,}000$ and $p = 32{,}000$; (**b**) impact of parameter $c_2$, wherein $c_1 = 40{,}000$, $\delta = 60{,}000$, $\xi = 30{,}000$ and $p = 32{,}000$; (**c**) impact of parameter $\delta$, $c_1 = 40{,}000$, $c_2 = 10{,}000$, $\xi = 30{,}000$ and $p = 32{,}000$; (**d**) impact of parameter $p$, $c_1 = 40{,}000$, $c_2 = 10{,}000$, $\delta = 60{,}000$ and $\xi = 30{,}000$ and (**e**) impact of parameter $\xi$, wherein $c_1 = 40{,}000$, $c_2 = 10{,}000$, $\delta = 60{,}000$ and $p = 32{,}000$.

### 5.2.2. Impact of Model Parameters on the Evolution Trend in Scenario 3

In scenario 3, to investigate the impact of monitoring cost ($c_1$) on the evolution trend, the initial point was assigned the value of (0.4, 0.6). Parameter $c_1$ ranged from 10,000 to 50,000 with a step size of 10,000, leaving all the other parameters at their default values. In the light of numerical simulation consequence as presented in Figure 5a, we found that when $c_1$ increased, the evolutionary rate of converging to the $A_1$ strategy adopted by the owner decreased. In other words, monitoring cost could slow down the probability of supervision on BIM applications.

Next, we further studied the impact of proprietary cost ($c_2$) on the selection of combined strategies adopted by two game players. The initial point was assigned the value of (0.4, 0.6). Figure 5b presents the simulation results, in which $c_2$ increased from 10,000 to 50,000 with a step size of 10,000. We could draw a conclusion that the evolutionary rate of converging to $B_1$ strategy adopted by the non-owner participant increased rapidly with the increase of parameter $c_2$.

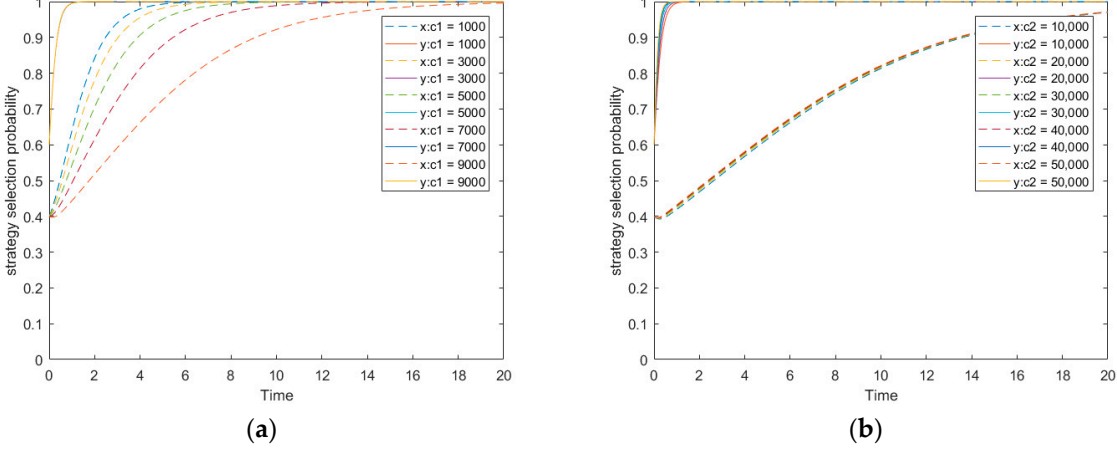

(**a**)

(**b**)

**Figure 5.** *Cont.*

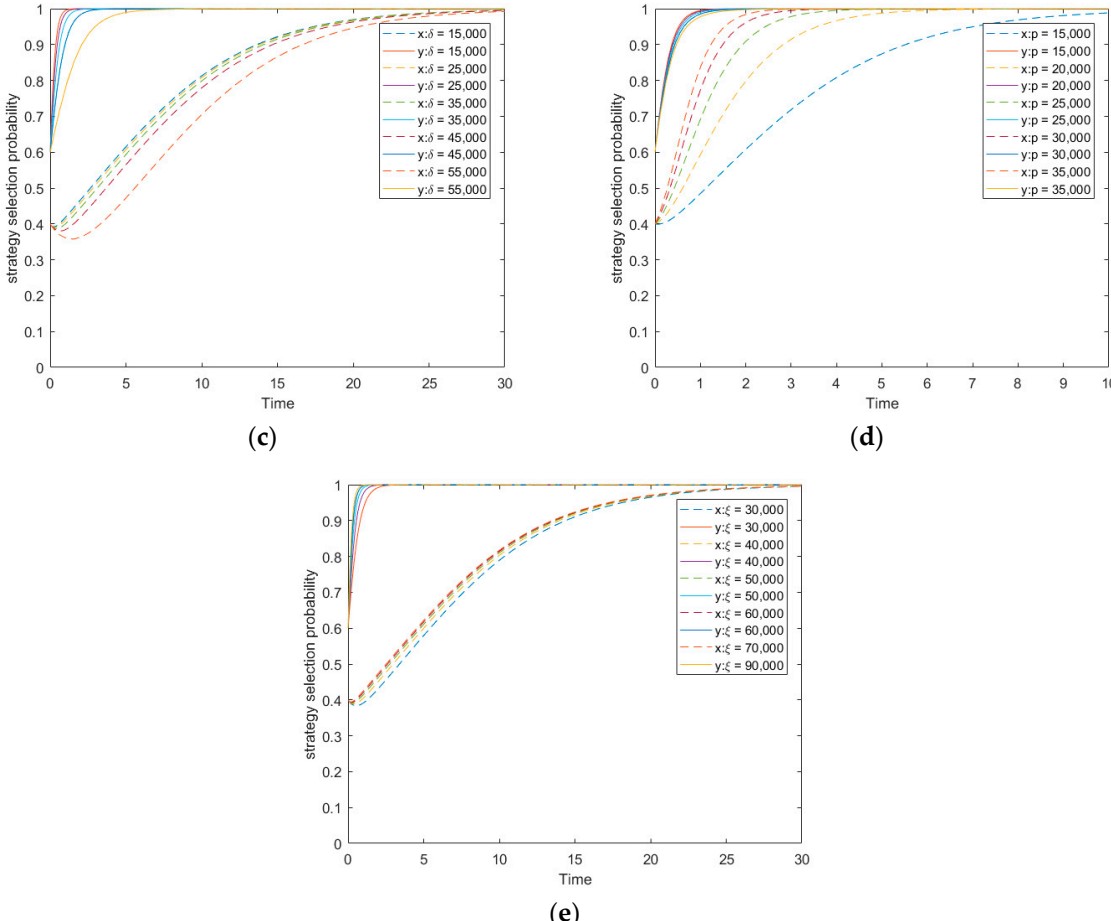

**Figure 5.** Impact of model variables on the evolution trend in scenario 3. (**a**) Impact of parameter $c_1$, wherein $c_2 = 10,000$, $\delta = 13,000$, $\xi = 58,000$ and $p = 12,000$; (**b**) impact of parameter $c_2$, wherein $c_1 = 10,000$, $\delta = 13,000$, $\xi = 58,000$ and $p = 12,000$; (**c**) impact of parameter $\delta$, wherein $c_1 = 10,000$, $c_2 = 10,000$, $\delta = 13,000$ and $p = 12,000$; (**d**) impact of parameter $p$, wherein $c_1 = 10,000$, $c_2 = 10,000$, $\delta = 13,000$ and $\xi = 58,000$ and (**e**) impact of parameter $\xi$, wherein $c_1 = 10,000$, $c_2 = 10,000$, $\delta = 13,000$ and $p = 12,000$.

Then, we analyzed the impact of incentive ($\delta$) on the evolutionary trend of BIM-based interactive behavior. The initial point was assigned the value of (0.4, 0.6). Parameter $\delta$ ranged from 15000 to 55,000 with a step size of 10,000. Based on simulation results, as shown in Figure 5c, we found that when the incentive increased, the evolutionary rate of converging to $B_1$ strategy decreased. That is to say, the incentive could effectively control the moral hazard behavior from the owner.

Subsequently, the impact of punishment ($p$) the evolutionary trend of BIM-based interactive behavior was analyzed. The starting point was set to (0.4, 0.6). Parameter $p$ ranged from 15,000 to 35,000 with a step size of 5000. From the simulation outcome, as shown in Figure 5d, we could find that if the owner strengthened the punishment, it could slow down the evolutionary rate of adopting $B_1$ strategy by the non-owner participant.

Lastly, we investigated the impact of speculative benefit ($\xi$) on the evolutionary direction. The starting point was set to (0.4, 0.6). Parameter $\xi$ ranged from 30,000 to 90,000 with a step size of 10,000. According to the simulation results as shown in Figure 5e we found that when speculative benefit increased, the probability of applying BIM by the non-owner participant at low level effort improved rapidly and eventually converged to 1.

### 5.2.3. Impact of Model Parameters on the Evolution Trend in Scenario 4

In scenario 4, to investigate the impact of five factors on evolutionary trend of combined strategies. The initial points were uniformly set as (0.4, 0.6), leaving all the other parameters unchanged when exploring the influence of a particular parameter. Specifically, monitoring cost ($c_1$) ranged from 2000 to 22,000 with a step size of 5000. Proprietary cost ($c_2$) ranged from 1000 to 9000 with a step size of 2000. Incentive ($\delta$) ranged from 6000 to 18,000 with a step size of 3000. Punishment ($p$) ranged from 50,000 to 90,000 with a step size of 10,000. Speculative benefit ($\xi$) ranged from 14,000 to 34,000 with a step size of 50,000. According to the simulation results, as shown in Figure 6a–e, an interesting finding could be obtained that no matter how the parameters change, the probability of adopting combined strategies displayed periodic recurrence over time, which means that the interactive behavior between two players changed dynamically, and would never reach the equilibrium.

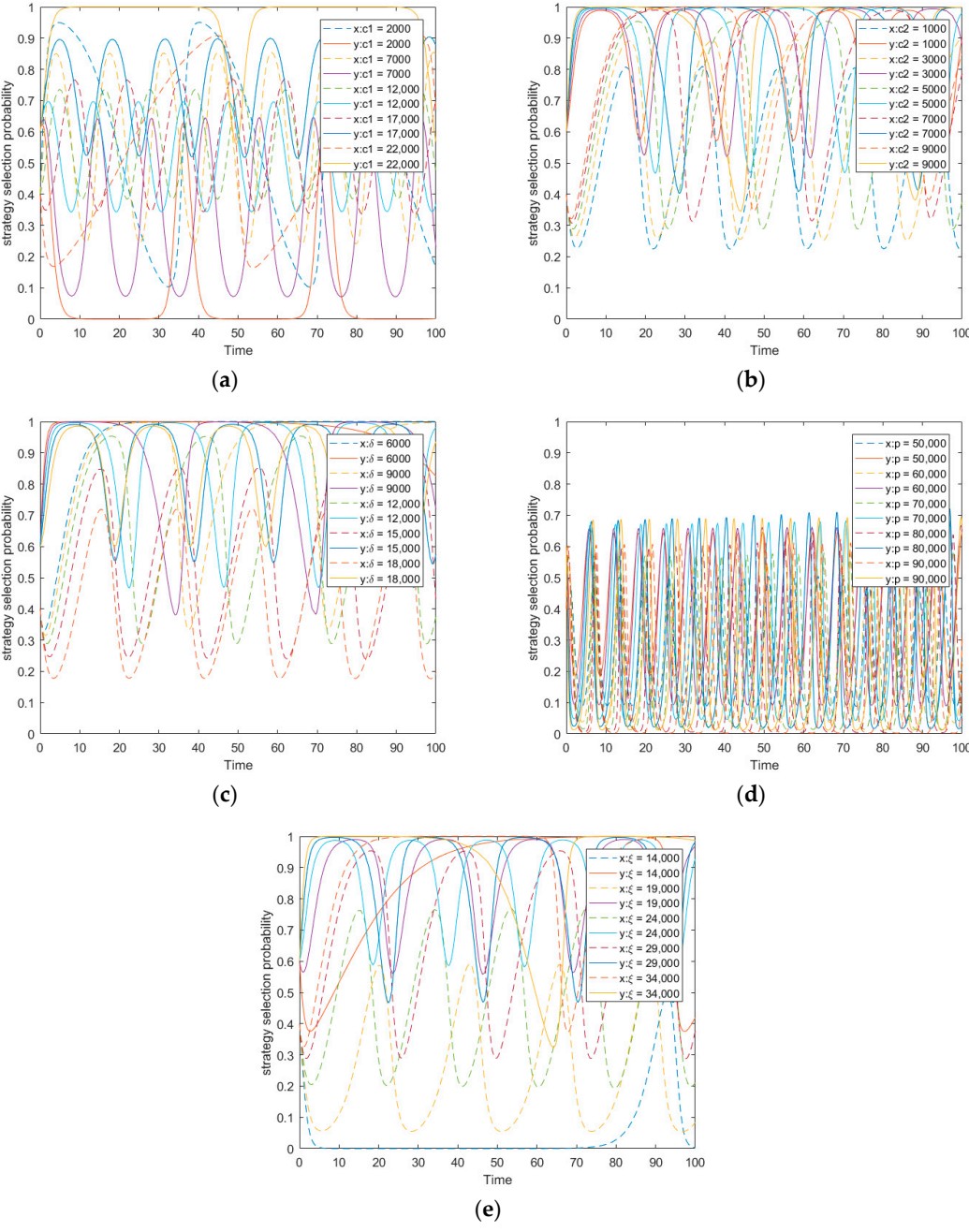

**Figure 6.** Impact of model variables on the evolution trend in scenario 4. (**a**) Impact of parameter $c_1$,

wherein $c_2 = 2000$, $\delta = 15{,}000$, $\xi = 26{,}000$ and $p = 23{,}000$; (**b**) impact of parameter $c_2$, wherein $c_1 = 20{,}000$, $\delta = 15{,}000$, $\xi = 26{,}000$ and $p = 23{,}000$; (**c**) impact of parameter, wherein $c_1 = 20{,}000$, $c_2 = 2000$, $\xi = 26{,}000$ and $p = 23{,}000$; (**d**) impact of parameter $p$, wherein $c_1 = 20{,}000$, $c_2 = 2000$, $\delta = 15{,}000$ and $\xi = 26{,}000$ and (**e**) impact of parameter $\xi$,wherein $c_1 = 20{,}000$, $c_2 = 2000$, $\delta = 15{,}000$ and $p = 23{,}000$.

### 5.2.4. Impact of Model Parameters on the Evolution Trend in Scenarios 5 and 6

In scenario 5, we first probed into the influence of monitoring cost ($c_1$) on the evolutionary trend of BIM-based strategy choice. The initial point was set to (0.4, 0.6). The value of parameter $c_1$ ranged from 20,000 to 60,000 with a step length of 10,000. Based on the simulation result that is presented in Figure 7a, with the increase of parameter $c_1$, the probability of implementing $A_1$ strategy rapidly converged to zero, which means that the owner would abandon supervision on BIM application rapidly when monitoring cost increases. The simulation result of $c_1$ in scenario 5 was consistent with the simulation result in scenario 6, as shown in Figure 8a, wherein parameter $c_1$ ranged from 31,000 to 193,000 with a step length of 40,000.

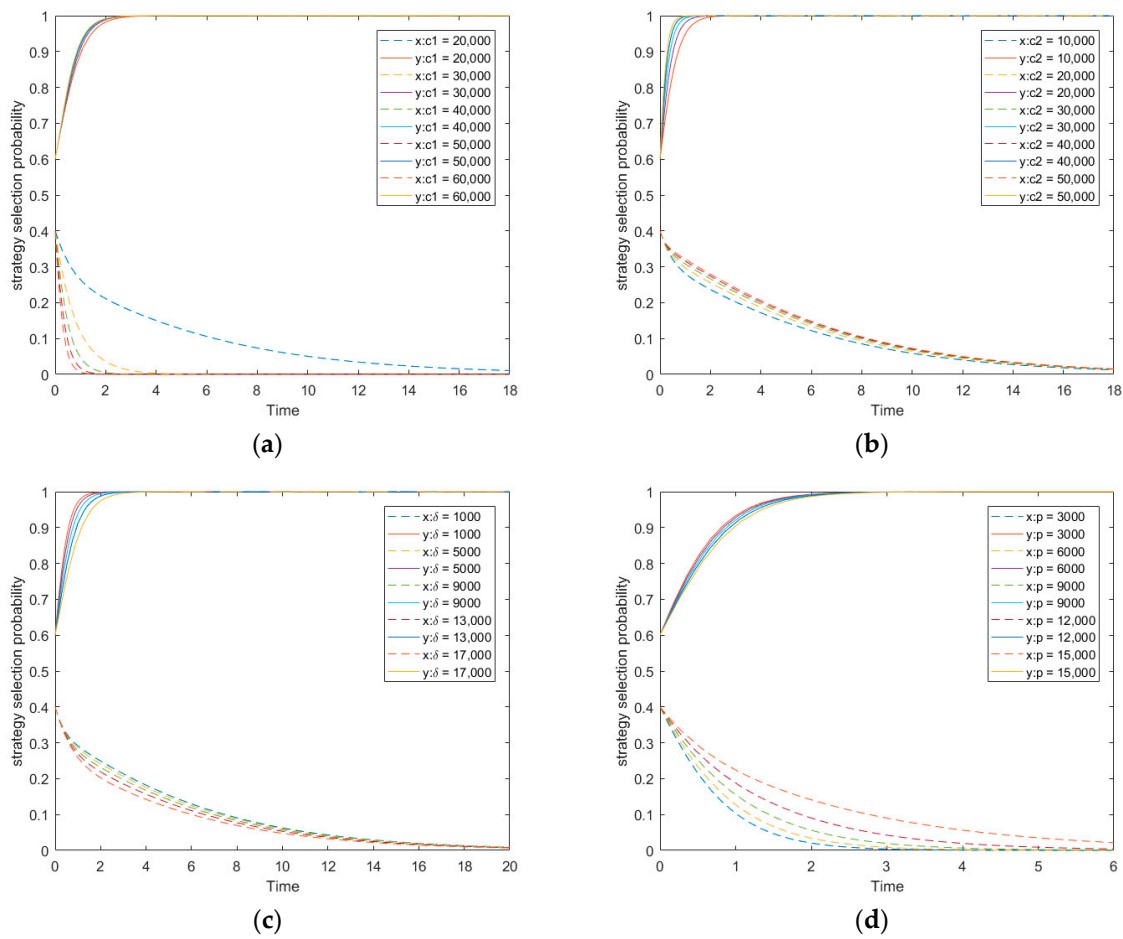

**Figure 7.** *Cont.*

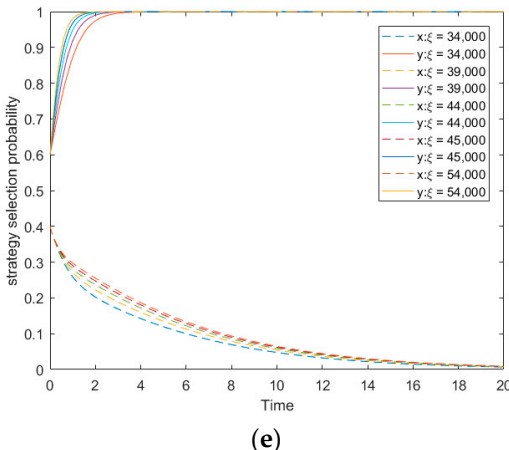

(**e**)

**Figure 7.** Impact of model variables on the evolution trend in scenario 5. (**a**) Impact of parameter $c_1$, wherein $c_2 = 2000$, $\delta = 15{,}000$, $\xi = 36{,}000$ and $p = 18{,}000$; (**b**) impact of parameter $c_2$, wherein $c_1 = 20{,}000$, $\delta = 15{,}000$, $\xi = 36{,}000$ and $p = 18{,}000$; (**c**) impact of parameter $\delta$, wherein $c_1 = 20{,}000$, $c_2 = 2000$, $\xi = 36{,}000$ and $p = 18{,}000$; (**d**) impact of parameter $p$, wherein $c_1 = 20{,}000$, $c_2 = 2000$, $\delta = 15{,}000$ and $\xi = 36{,}000$ and (**e**) impact of parameter $\xi$, wherein $c_1 = 20{,}000$, $c_2 = 2000$, $\delta = 15{,}000$ and $p = 18{,}000$.

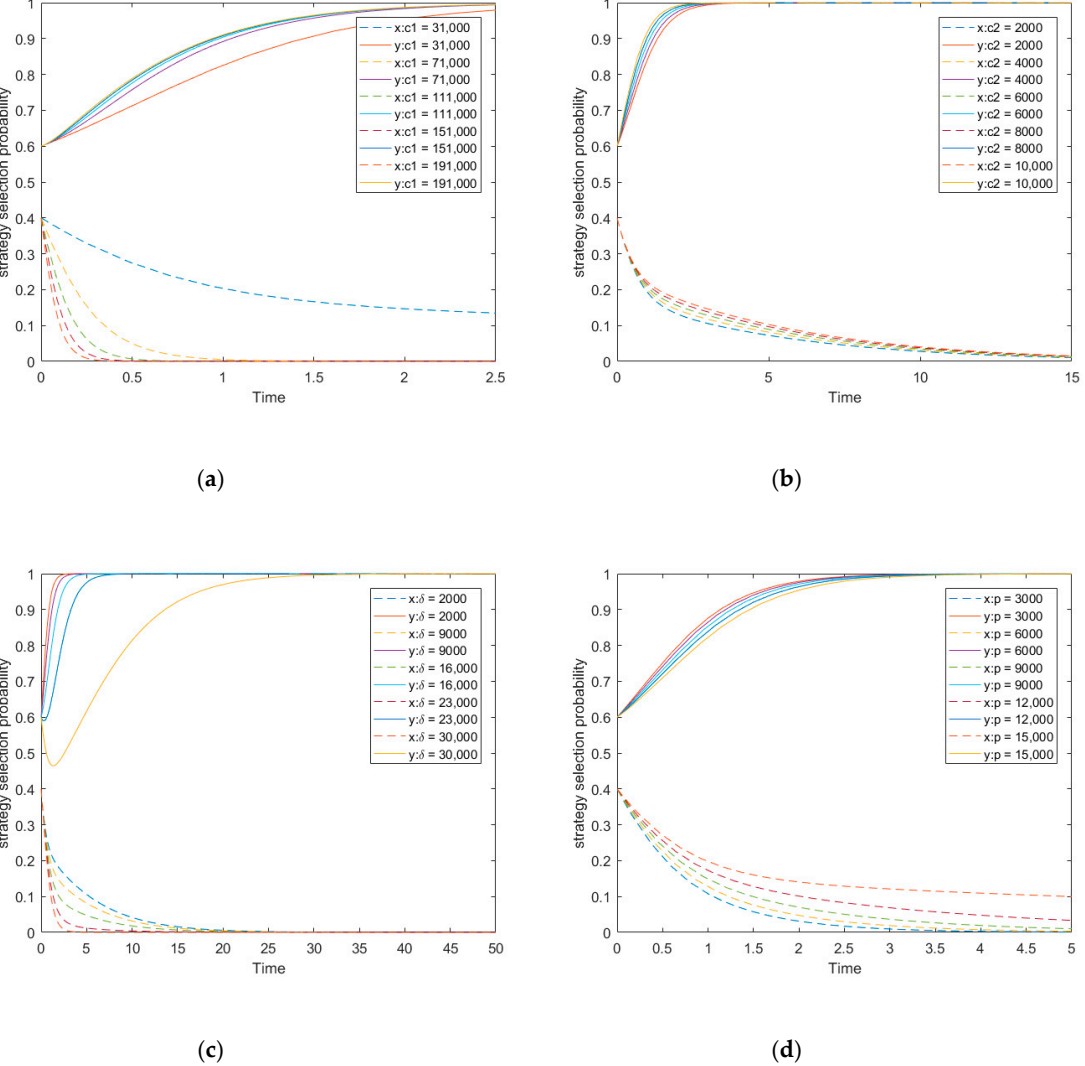

(**a**)

(**b**)

(**c**)

(**d**)

**Figure 8.** *Cont.*

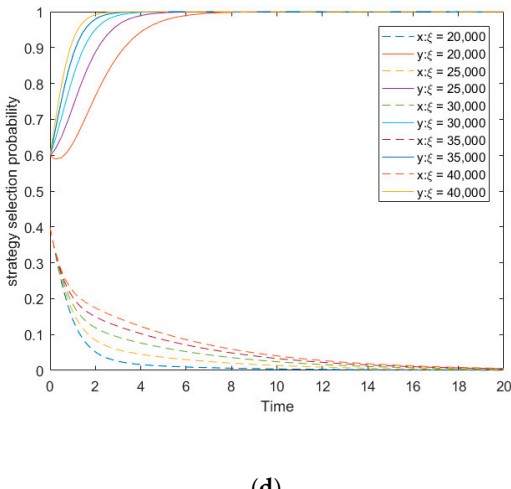

(**d**)

**Figure 8.** Impact of model variables on the evolution trend in scenario 6. (**a**) Impact of parameter $c_1$, wherein $c_2 = 1000$, $\delta = 12,000$, $\xi = 31,000$ and $p = 30,000$; (**b**) impact of parameter $c_2$, wherein $c_1 = 32,000$, $\delta = 12,000$, $\xi = 31,000$ and $p = 30,000$; (**c**) impact of parameter $\delta$, wherein $c_1 = 32,000$, $c_2 = 1000$, $\xi = 31,000$ and $p = 30,000$; (**d**) impact of parameter $p$, wherein $c_1 = 32,000$, $c_2 = 1000$, $\delta = 12,000$ and $\xi = 31,000$ and (**e**) impact of parameter $\xi$, wherein $c_1 = 32,000$, $c_2 = 1000$, $\delta = 12,000$ and $p = 30,000$.

Next, we explored the impact of proprietary cost ($c_2$) on the evolutionary trend in scenario 5. The initial point was assigned the value of (0.4, 0.6). The value of parameter $c_2$ ranged from 10000 to 50,000 with a step length of 10,000. Based on the simulation result as presented in Figure 7b, we found that with the increase of parameter $c_2$, the evolutionary rate of adopting $B_1$ strategy increased rapidly. That is, the probability of moral hazard behavior incurred by the non-owner participant rapidly converged to 1, eventually delivering poor quality BIM-based products to the owner. The simulation result of $c_2$ in scenario 5 was consistent with the simulation result in scenario 6, as shown in Figure 8b, wherein parameter $c_2$ ranged from 10,000 to 50,000 with a step length of 10,000.

Then, we investigated the impact of incentive ($\delta$) on the evolutionary trend in scenario 5. The initial point was assigned the value of (0.4, 0.6). The value of parameter $\delta$ ranged from 1000 to 17,000 with a step length of 10,000. Based on the simulation result as presented in Figure 7c, increasing of $\delta$ extends the evolutionary time of adopting evolving to $B_1$ strategy. In other words, the incentive could effectively control the moral hazard behavior. The simulation result of parameter $\delta$ in scenario 5 was consistent with the simulation result in scenario 6, as shown in Figure 8c, wherein parameter $\delta$ ranged from 1000 to 17,000 with a step length of 4000.

To investigate the impact of punishment ($p$) on the strategy selection, the initial point was assigned the value of (0.4, 0.6). The value of parameter $p$ ranged from 3000 to 15,000 with a step length of 3000. According to simulation outcome as shown in Figure 7d, we found that when parameter $p$ increased, the evolutionary rate of employing $A_1$ strategy decreased, which means that the punishment was helpful for slowing down the motivation of applying BIM at low level effort by the non-owner participant. The simulation result of parameter $p$ in scenario 5 was consistent with the simulation result in scenario 6, as shown in Figure 8d wherein parameter $p$ ranged from 3000 to 15,000 with a step length of 3000.

Last, we analyzed the influence of speculative benefit ($\xi$) derived from the moral hazard behavior of the non-owner participant on evolutionary trend. The initial point was assigned the value of (0.4, 0.6). The value of parameter $\xi$ ranged from 34,000 to 54,000 with a step length of 5000. We found that the non-owner participant would likely applying BIM at low level effort under the temptation of high speculative benefit, as shown in Figure 7e. The simulation result of parameter $\xi$ in scenario 5 was consistent with the simulation result in scenario 6, as shown in Figure 8e, wherein parameter $\xi$ ranged from 34,000 to 54,000 with a step length of 5000.

## 6. Results and Discussion

We verified the proposed evolutionary game model through detailed numerical simulations. These simulation results manifested the novel model in this study could effectively and accurately capture the interactive behavior of BIM-based strategies selection among integrated project team members, and thus successfully predicted the evolutionary trend with the changes of model parameters. The proposed novel model, different from the traditional game model, extends the limitations that game players should act rationally and hold complete information of the competitors. Thus two significant characteristics, namely bounded rationality of game players and incomplete information of strategy choosing, were introduced into the proposed novel-innovative model. In this context, the simulation results would be consistent with the real situation. Based on the simulation results, some valuable findings could be acquired to help integrated project managers develop related countermeasures for the purpose of controlling moral hazard behavior and improving performance of an integrated project team.

(1) Based on the evolutionary paths, as presented in Figure 2a,b, we discovered that when the incentive payment provided by the owner (i.e., player 1) can cover the speculative benefit and proprietary cost incurred by the non-owner participant (i.e., player 2), namely $\delta > \xi + c_2$, both game players would choose the combined strategy $(A_2, B_2)$. Thus, player 2 would be more likely to apply BIM at high level effort, and work hardly to deliver a BIM outcome with high quality. Moreover, the non-owner participant would try their best to explore meritorious application of BIM, which may bright in a more significant added value for the overall project. Against this background, the owner relaxes supervision and is in favor of BIM exploration. Choosing a moral hazard behavior or not is identified as a function of cost and benefit. Nevertheless, proprietary cost derived from high level effort in using BIM and speculative benefit from moral hazard behavior is often ignored when analyzing the impact of cost on moral hazard behavior in IPD-based projects in extant literatures, which easily leads to an unreliable conclusion that may cloud the judgment of the project manager. Compared with previous researches, we demonstrated that proprietary cost and speculative benefit affected significantly the selection of behaviors.

(2) When the incentive payment provided by the owner cannot cover the speculative benefit and proprietary cost incurred by the non-owner participant, and the punishment is greater than the monitoring cost and less than the sum of speculative benefit and proprietary cost minus the incentive payment, namely $\delta - c_2 < \xi, c_1 < p < \xi + c_2 - \delta$, the evolution path as shown in Figure 1c, both game players eventually selected the combined strategy $(A_1, B_1)$. Player 2 chose to use BIM at low level effort, and only provided conceptual BIM-based products to player 1. The owner would strengthen supervision and strictly stipulated the scope of BIM application to regulate the behavior of player 2. When we continued to decrease the punishment and made it less than the monitoring cost, two kinds of situation existed, namely, (1) $\delta - c_2 < \xi, p < c_1$ and $p < \xi + c_2 - \delta$ and (2) $\delta - c_2 < \xi, \xi + c_2 - \delta < p < c_1$. Based on evolution paths as shown in Figure 1e,f, we could find the combined strategy changes from $(A_1, B_1)$ to $(A_2, B_1)$. Therefore, player 1 would relax supervision on BIM application and detailed content of work. Compared to traditional studies that only highlight the influence of incentive mechanism on the agent within the principal-agent framework, the simulation results in this study demonstrated that the owner was also influenced by punishment that was rarely paid attention to in extant literatures.

(3) To the best of our knowledge, very few studies have successfully explained the confused phenomenon of why BIM application is in a chaos state from the theory aspect. The evolution path, as shown in Figure 2d, indicates that the proposed model in this study had no equilibrium stable strategy when the incentive payment provided by the owner could not cover the speculative benefit and proprietary cost incurred by the non-owner participant, and the punishment was greater than the monitoring cost, meanwhile the punishment was larger than the sum of speculative benefit and proprietary cost minus the incentive payment, namely $\delta - c_2 < \xi, p > c_1$ and $p > \xi + c_2 - \delta$. In this context, the simulation results as presented in Figure 5a–e, indicate that with the players'

strategies selection changing frequently, the uncertainty of integrated project team increased greatly. Moreover, it would consume a lot of organizational resources, including budget, time and energy to deal with communication and technology problems, thereby leading to organizational productivity declines and project failure.

(4)　By exploring the influences of special parameters on the evolutionary trend, some significant findings could be acquired. First of all, according to the simulation results as shown in the subgraph (a) of Figures 3–5, 7 and 8, we found that when the monitoring cost increased, the evolutionary rate of adopting $A_1$ strategy by the owner was slowed. On this background, the owner would likely lose supervision on BIM application and exploration with the increase of the monitoring cost. Secondly, according to the simulation results as presented in the subgraph (b) of Figures 3–5, 7 and 8, we could see that the moral hazard behavior from the non-owner participant was influenced positively, which means that the larger the proprietary cost, the more motivation the non-owner participant would adopt to use BIM at the low level effort. Thirdly, based on the simulation results as shown in the subgraph (c) of Figures 3–5, 7 and 8, the incentive payment provided by the owner had negative correlations with the occurring probability of the non-owner participants' moral hazard behavior. When the incentive payment increased and reached a certain level, the willingness of applying BIM at low level effort by the non-owner participant decreased and ultimately gave up moral hazard behavior. Fourthly, from numerical simulation results, as shown in the subgraph (d) of Figures 3–5, 7 and 8, we could see that the larger the parameter of punishment, the slower the evolutionary rate of adopting BIM by the non-owner participant at low level effort, which means that the punishment mechanism played an important role in controlling moral hazard behavior from the non-owner participant. Lastly, speculative benefit, according to simulation results as shown in the subgraph (e) of Figures 3–5, 7 and 8, significantly affected the strategies selection. If it was large enough, the non-owner participant would adopt moral hazard behavior without hesitation, even though they were fined by the owner.

## 7. Conclusions and Implications

The IPD is a delivery system that has received widespread attention of global researchers and enterprises. However, the IPD system has not yet shifted from single pilot or particular-purposed cases to large-scale applications. In the IPD system, BIM's huge advantages are far from being exploited, which directly leads to the delivered outcomes below expectations, thereby causing obstacles to widespread application of the IPD system. All kinds of elements that lead to the dilemma of IPD applications have been studied by researchers, such as technology maturity, knowledge sharing mechanism, trust relationship, culture environment, etc. [41–44]. However, very few studies have focused on the moral hazard behavior that is a very important factor resulting in the failure of BIM applications in an integrated project team. The objective of this paper was to explore the evolutionary mechanism of moral hazard behavior from the non-owner participant in the integrated project team. We proposed a novel model to capture dynamically the behavior of BIM-based strategy selections using evolutionary game theory. The simulation experiments conducted with MATLAB 2016a demonstrated that when incentive payment was higher than the sum of speculative benefit and proprietary cost, interactive behavior of both game players would move toward the optimal portfolio strategy. There was a strong negative correlation between incentive payment and punishment imposed by the owner and the motivation of hazard behavior, which indicates that the larger the incentive payment, the smaller the probability of adopting BIM at low level effort. On the contrary, speculative benefit and proprietary cost affected positively the implementation probability of moral hazard behavior of BIM application, revealing that the two factors played the key role in inducing moral hazard behavior of employing BIM. In term of simulation results, some valuable implications acquired could be provided for the integrated project leader and competent departments of construction administration to develop measures, for the sake of controlling moral hazard behavior and improving project performance.

(1) For integrated project team leaders, the priority is to make sure that the incentive payment is higher than the sum of speculative benefit and proprietary cost incurred by the non-owner participant when developing an incentive mechanism. Under this situation, the incentive mechanism can effectively restrain the non-owner participants' moral hazard behavior, reduce the owner's supervision cost and improve the overall project benefit, which is the best choice for all stakeholders. Whether the owner strengthens supervision or not on BIM application, the non-owner participant would always use BIM at high level effort. Therefore, it is not hard to see that the incentive plays a dominant role in making the non-owner participant apply BIM at high level effort. However, the project manager should not just depend on the incentive mechanism. The punishment mechanism that has a strong deterrent effect on moral hazard behavior should also be considered, when certain constraint conditions are satisfied. In fact, compared to a single incentive mechanism, a more significant implementation effect on controlling moral hazard behavior can be achieved under both the incentive and punishment mechanism.

(2) Speculative benefit is the main motivation for the non-owner participant to use BIM at low level effort. Therefore, reducing speculative benefit is a very important way to control moral hazard behavior. We have to bring in the third party, namely the competent department of construction administration, for the purpose of solving this problem from the perspective of the overall market environment. To prompt widespread application and healthy development of the integrated project delivery system, the competent departments of construction administration should establish the information sharing platform and credit evaluation system, improve the information dissemination mechanism. For example, if the non-owner participants adopt moral hazard behavior, their credit rating would be downgraded or even forbidden to enter the construction market. In this context, the violation cost would be increased greatly, and thus speculative benefit expected by the non-owner participant can be offset. Consequently, the moral hazard behavior can be controlled effectively from the external environment.

(3) Proprietary cost should be paid more attention to. According to the simulation results, there is a positive correlation between the proprietary cost and probability of occurrence of moral hazard behavior. In order to reduce proprietary cost incurred by the non-owner participant, the integrated project team manager should first conduct reengineering of the business process that significantly affects integration between the IPD system and BIM. The traditional work process that is established based on computer-aided design has not adapted to deep application of BIM, which increases implementation cost [13]. Moreover, the project manager should check and ensure the BIM software adopted by each stakeholder compatible, and provide the uniformed data standard manual to make information flow unimpeded, for the purpose of reducing technology cost [45]. Last but not least, the integrated project team leader should select integrated project team members with rich experiences in BIM-and IPD-based projects, so as to reduce trail-and-error cost that is also believed as an important kind of proprietary cost [46].

This study has both theoretical and practical contributions. In theoretical aspects, the interesting findings obtained could help us make clear the evolutionary mechanism of moral hazard behavior of BIM applications from the non-owner participant, and broaden our knowledge of relationships between IPD system operations and BIM. In practice, simulation results in different parametric intervals could help integrated project team managers and competent departments of construction administration develop reasonable measures to control moral hazard behavior of BIM applications, and thus prompt widespread application and healthy development of the IPD system in the construction market.

There are some limitations that should be further studied in the future research. First of all, practical cases of IPD are hard to get in China, the interesting findings are obtained primarily from simulation results, which may leave out some useful information that is rarely mentioned in prior studies. In the following research, the authors would try their best to collect and analyze practical cases for the purpose of validating the findings in this study and further supplement them. Secondly, how to quantify monitoring cost, speculative benefit and proprietary cost have not been discussed

sufficiently in this paper, which is a definitely complicated systematic project that needs to consider manifold factors to analyze and calculate. This may influence the development of incentive and punishment mechanism for the project manager in practice. In future research, the authors will find an effective and systematic way to quantify them.

**Author Contributions:** Conceptualization, Y.D.; Methodology, Y.D. and H.Z.; Software, Y.D. and H.Z.; Validation, H.X. and Y.Y.; Formal Analysis, H.X.; Investigation, Y.D. and H.Z.; Resources, Y.Y.; Data Curation Y.D.; Writing-Original Draft Preparation, Y.D.; Writing-Review Editing, Y.D. and Y.Y.; Visualization, Y.D. and H.Z.; Supervision, H.Z.; Project Administration, Y.Y.

**Acknowledgments:** The authors gratefully acknowledge those experts for their constructive comments and suggestions.

**Conflicts of Interest:** The authors declare no conflict of interest.

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
