# Peer review of "Exploring the Moral Hazard Evolutionary Mechanism for BIM Implementation in an Integrated Project Team"

_sustainability, doi:10.3390/su11205719_

Round 1

Reviewer 1 Report

Ms. Ref. No. : Sustainability-611433 Title: Exploring the Moral Hazard Evolutionary Mechanism for BIM Implementation in Integrated Project Team Reviewer' comments: First of all, I would like to thank authors for submitting their manuscript to Journal of Sustainability. The authors presented an analysis of the interactions between moral hazard behavior and IPD system by numerical simulations. The manuscript is acceptable for publication.

Author Response

thank you very much for reviewing this paper and affirming  the authors' research work. 

Reviewer 2 Report

In many ways this is an insightful paper and approach. The subject that is not covered and is central to this paper is Target Value Design. Discussion about to Target Value Design should be added to ensure a more comprehensive view and understanding of IPD is provided. Also more could have been mentioned about the BIM Lean correlations.

Author Response

Dear reviewer,

thank you very for your significant comments, the authors have replied to your suggestions point by point, the detailed reply to comments is upload as a word file.

Round 2

Reviewer 2 Report

The inclusion of TVD gives a more holistic understanding of the subject area.